# Precursor B-ALL Cell Lines Differentially Respond to SYK Inhibition by Entospletinib

**DOI:** 10.3390/ijms22020592

**Published:** 2021-01-08

**Authors:** Sina Sender, Anett Sekora, Simon Villa Perez, Oleksandra Chabanovska, Annegret Becker, Anaclet Ngezahayo, Christian Junghanss, Hugo Murua Escobar

**Affiliations:** 1Division of Medicine, Department of Hematology, Oncology and Palliative Medicine, University of Rostock, 18057 Rostock, Germany; sina.sender@med.uni-rostock.de (S.S.); Anett.Sekora@med.uni-rostock.de (A.S.); dsibsimonvilla@gmail.com (S.V.P.); oleksandra.chabanovska@uni-rostock.de (O.C.); Christian.Junghanss@med.uni-rostock.de (C.J.); 2Department of Cell Physiology and Biophysics, Institute of Cell Biology and Biophysics, Leibniz University Hannover, 30419 Hannover, Germany; becker@cell.uni-hannover.de (A.B.); ngezahayo@cell.uni-hannover.de (A.N.)

**Keywords:** entospletinib, Ento, GS-9973, B-ALL, acute lymphoblastic leukemia, expression analysis, SYK, BCR, pathway-specific inhibitors

## Abstract

Background: Impaired B-cell receptor (BCR) function has been associated with the progress of several B-cell malignancies. The spleen tyrosine kinase (SYK) represents a potential therapeutic target in a subset of B-cell neoplasias. In precursor B-acute lymphoblastic leukemia (B-ALL), the pathogenic role and therapeutic potential of SYK is still controversially discussed. We evaluate the application of the SYK inhibitor entospletinib (Ento) in pre- and pro-B-ALL cell lines, characterizing the biologic and molecular effects. Methods: SYK expression was characterized in pre-B-ALL (NALM-6) and pro-B-ALL cell lines (SEM and RS4;11). The cell lines were exposed to different Ento concentrations and the cell biological response analyzed by proliferation, metabolic activity, apoptosis induction, cell-cycle distribution and morphology. BCR pathway gene expression and protein modulations were further characterized. Results: Ento significantly induced anti-proliferative and pro-apoptotic effects in NALM-6 and SEM, while barely affecting RS4;11. Targeted RNAseq revealed pronounced gene expression modulation only in NALM-6, while Western Blot analyses demonstrated that vital downstream effector proteins, such as pAKT, pERK, pGSK3β, p53 and BCL-6, were affected by Ento exposure in the inhibitor-sensitive cell lines. Conclusion: Different acting modes of Ento, independent of pre-BCR dependency, were characterized, unexpected in SEM. Accordingly, SYK classifies as a potential target structure in a subset of pro-B-ALLs.

## 1. Introduction

B-cell receptor (BCR) signaling, as an essential regulating network, provides the main signals for survival and proliferation in hematopoietic cells [1]. In B-lymphoid malignancies, BCR signaling can be impaired by activating mutations, deletions or hyper-activated tonic BCR signaling. Impaired function of the BCR can therefore lead to B-cell malignant pathogenesis as chronic lymphocytic leukemia (CLL) or lymphoma [2]. Based on this, the BCR signaling cascade with its involved genes can serve as target genes for targeted therapies in B-cell neoplasias as CLL and B-cell lymphomas. A potential target structure is represented by SYK, the spleen tyrosine kinase. SYK, as a 72 kDa, non-receptor tyrosine kinase proximal to the BCR, is one major checkpoint for B-cell maturation and required for pro-B- (cytoplasmic Igµ^−^/surface IgM^−^) to pre-B-cell (cytoplasmic Igµ^+^/surface IgM^−^) transition [3,4,5,6]. During immunoreceptor engagement, the cytoplasmic tails of the BCR auxiliary transmembrane proteins Ig-α and Ig-β, which constitute the immunoreceptor tyrosine-based activation motifs (ITAMs), are phosphorylated [7]. As a result, SYK is recruited to the ITAMs, becomes phosphorylated, and thus activated [2]. Now, SYK directly phosphorylates BLNK and transmits signals to several signaling pathway cascades downstream of the BCR, including PI3K/AKT, ERK and BTK/PLCγ2 [8,9,10,11,12]. Moreover, SYK fulfills certain functions independent of BCR signaling, e.g., signal transduction from tumor necrosis factor (TNF) superfamily receptors, integrins and cytokines [2]. Activation of these pathways leads to proliferation, differentiation, cytoskeletal remodeling, cytokine release and survival [2]. In several hematological malignancies, such as CLL [13], diffuse large B-cell lymphoma (DLBCL) [14], mantle cell lymphoma (MCL) [15] and follicular lymphoma (FL) [16], SYK acts as a proto-oncogene and is thus involved in tumorigenesis. Here, tonic BCR signaling leads to constitutively activated SYK [17].

Indeed, Goodman et al. reported a SYK deficiency in pediatric precursor pro-B-ALL [18], whereas Perova et al. demonstrated in pediatric and adult B-ALL samples a pre-BCR independent SYK activation in high-risk precursor B-ALL patients and with that an apoptosis induction by specific SYK inhibition [19].

Based on pre-clinical evaluation, several SYK inhibitors are developed and under clinical investigation, e.g., fostamatinib disodium (R788) [20], entospletinib (GS-9973) [21] and cerdulatinib (PRT062070) [22]. In 2018, FDA approval of fostamatinib disodium ensued as Tavalisse for patients with chronic immune thrombocytopenia with prior unsuccessful treatment [23]. A second-generation compound, entospletinib (GS-9973), as one of the most selective SYK inhibitors, was developed as a consequence of the prodrug fostamatinib, which showed dose-limiting adverse events in patients with relapsed or refractory B-cell lymphoid malignancies, probably based on poor kinase selectivity and off-target effects [20]. Entospletinib is an ATP competitive kinase inhibitor and disrupts the kinase activity of SYK. Additionally, entospletinib exhibits an excellent selectivity profile with high efficacy for SYK inhibition and presumably lower dose-limiting adverse effects [24]. Especially, the combination of PI3Kδ inhibitor idelalisib and entospletinib showed to disrupt the BCR and chemokine signaling in CLL pre-clinically [25]. A Phase 2 study with CLL patients also showed lymph node reduction and partial response with acceptable tolerability by entospletinib (Ento) administration [21].

The observed effects based on SYK inhibition in precursor B-ALL in vitro are mostly defined by BCR or pre-BCR dependency. Köhrer et al. likewise showed an anti-tumorigenic potential of SYK inhibition by PRT318 in pre-B-ALL cell lines (BCR^+^), whereas pro-B-ALL cell lines were not affected (BCR^−^) [4].

Due to the controversially discussed SYK function, especially in precursor B-ALL, the characterization of SYK as a potential target structure is an essential need. Here, we provide an in vitro-based study, which aimed to illustrate the application of entospletinib as a specific SYK inhibitor on pre- and pro-B-acute lymphoblastic leukemia cell lines, in order to further demonstrate a possible application in a subset of precursor B-ALL and, moreover, to elucidate the underlying diverging acting modes.

## 2. Results

### 2.1. Pro-B-ALL Cell Line SEM is Characterized by High SYK and pSYK Expression

SYK expression was evaluated by Western blot analyses, immunofluorescence staining and intracellular flow cytometry. Pre-B-ALL cell line NALM-6 (pre-BCR^+^) revealed the highest amount of total SYK during Western blot analyses (Figure 1a). Pro-B-ALL cell lines SEM and RS4;11 (pre-BCR^−^) showed less total SYK expression. Total SYK expression was also visualized by immunofluorescence staining in all cell lines and the strongest signal was confirmed in the pre-B-ALL cell line NALM-6 within all tested B-ALL cell lines (Figure 1b). The positive-control B-cell lymphoma cell line SU-DHL-4 revealed a high SYK expression, while the negative-control T-cell lymphoma cell line SUP-T1 was SYK-negative (Figure 1b).

Furthermore, intracellular flow cytometry demonstrated almost entirely SYK-positive cells for all tested cells except for the negative control SUP-T1. Comparison of the mean fluorescence intensities (MFIs) revealed significant differences between the B-ALL cell lines showing highest MFI values for NALM-6.

In contrast to the total form of SYK, the amount of the activated and phosphorylated SYK form (phospho SYK; pSYK) varied within the analyzed cell lines using Western blot detection (Figure 1a). Two different SYK phosphorylation sites were analyzed for pSYK characterization via Western blot. Thereby, the auto-phosphorylation site Tyr525/526 and activation site Y352 (Interdomain Linker Region [26]) revealed an unexpected high amount of activated SYK in the pro-B-ALL cell line SEM and less in NALM-6 and RS4;11 (Figure 1a). Flow cytometric detection of pSYK, using an alternative antibody detecting phosphorylation site Y348, revealed low pSYK levels for all cell lines except for the IgG-stimulated, positive-control cell line SU-DHL-4 (Figure 1c). Percentagewise, again the pro-B-ALL cell line SEM revealed the highest pSYK level within the unstimulated B-ALL cell lines.

### 2.2. Basal Expression Analysis of Key B-Cell Receptor Genes in the B-Acute Lymphoblastic Cell Lines NALM-6, SEM and RS4;11

To evaluate the basal RNA expression level of SYK as well as key regulators in the B-cell-receptor pathway, we performed targeted RNAseq with 179 genes involved in the BCR, PI3K/AKT pathway signaling and more. The highest SYK expression was detected in pre-B-ALL NALM-6 with two-fold higher expression compared to both the pro-B-ALL cell lines (the normalized read counts of a subset of vital kinases and proteins of these pathways are shown in Figure 2 and Appendix A, and the read counts and normalized read counts in Appendix A).

*AKT1* and *AKT2* showed similar expression in all three cell lines, while *AKT3* is highly expressed in both the pro-B-ALL cell lines and only slightly expressed in the pre-B-ALL cell line NALM-6. The *BCL-2* and *BCL-6* expression differs between the cell lines, whereas *BCL-6* is highest expressed in RS4;11. Furthermore, *BTK* was found present in all three cell lines at a similar kind of intensity. The PI3K subunits *PI3Kα* and *PI3Kβ* are likely moderately expressed in all three cell lines, whereas the *PI3Kδ*, *PI3Kγ* and *PI3KR1* subunits vary between the cell lines. *PI3Kδ* is highest expressed in NALM-6; the merest *PI3Kγ* expression is revealed in SEM and the *PI3KR1* subunit is significantly highly expressed in NALM-6. *TP53* and *ERK* expression is nearly equal in all cell lines. All MAP kinases are represented in all cell lines in a similar intensity. However, *PTPN6* revealed the highest expression in the pro-B-ALL cell line SEM.

### 2.3. Ento Sensitivity Is Independent of the Pre-BCR Status

Ento exposure on pre-B-ALL NALM-6 (pre-BCR^+^) showed a concentration and slightly time-dependent decrease in cell proliferation, most affected at 72 h (Figure 3a). The metabolic activity was only moderately affected by Ento and re-increases with a longer incubation time. Interestingly, Ento exposure showed anti-proliferative effects also on the pro-B-ALL cell line SEM (pre-BCR^−^) (Figure 3b). Significant effects were observed starting with 5 µM for 24 h and 1 µM Ento for 48 h and 72 h exposure. The metabolic activity was similarly affected, slightly ascending with incubation time. In contrast, the pro-B-ALL RS4;11 (pre-BCR^−^) showed only a moderate reduction in proliferation at the highest concentration 10 µM and 20 µM applied for 48 h and 72 h (Figure 3c). The IC50 calculation based on 72 h proliferation is shown in Appendix A (Appendix A), where the IC50 of RS4;11 was not reached.

### 2.4. Pro-Apoptotic Effects Are Provoked by High Entospletinib Concentrations, Independent of Pre-BCR Dependency

High Ento concentrations induced significant early and late apoptosis/necrosis in the pre-B-ALL NALM-6 cell line during all tested time points (Figure 4a). However, in the pro-B-ALL cell line SEM, high Ento concentrations induced significant early and late apoptosis/necrosis to a higher extent even compared to pre-B-ALL NALM-6. During the early time point (24 h), Ento mainly provokes late apoptosis, while the early apoptosis rate increased during the later time points (48 h and 72 h) (Figure 4b). In contrast, pro-B-ALL RS4;11 was almost resistant to apoptosis induction by Ento (Figure 4c). Exemplarily, apoptosis measurements of the SEM cells are given in Appendix A (Appendix A).

### 2.5. Morphological Characterization Indicates Pro-Apoptotic Effects in Pre- and Pro-B-ALL Cell Lines In Vitro

The representative images of the Ento-exposed B-ALL cells (Figure 4d) indicate morphological changes in the sensitive B-ALL cell lines and moderate changes in the non-responsive cell line RS4;11 cells after Ento exposure. In NALM-6, cellular fragmentation and nuclear vacuolization can be determined. The light microscopy images of SEM indicate membrane blebs, nuclear vacuolization, cellular fragmentation and slight chromatin condensation after 72 h Ento exposure. The observed morphological changes indicate stress and apoptosis induction in both the sensitive B-ALL cell lines. In RS4;11, mainly membrane blebs and nuclear vacuolization were observed. Representative images of all the tested concentrations (1 µM, 5 µM and 10 µM) and all time points (24 h, 48 h and 72 h) further confirmed morphological changes in all B-ALL cell lines and indicates apoptosis induction (Appendix A).

### 2.6. Entospletinib Does Not Induce Cell-Cycle Arrest in B-ALL Cell Lines

Cell-cycle distribution was investigated in the pre-B-ALL cell line NALM-6 and pro-B-ALL cell lines SEM and RS4;11 after Entospletinib exposure. Both concentrations of Ento (low concentration 1 µM or high concentration 10 µM) were not able to induce changes in cell-cycle distribution after 72 h incubation in the tested B-ALL cell lines (Figure 5a–c). Exemplarily, the cell-cycle distribution of the Ento-exposed SEM cells is shown in Figure 5d.

### 2.7. Entospletinib Downregulates Gene Expression of PTPN6 and BCL-6 in Pre-B-ALL NALM-6

Principle component analysis (PCA) indicates clear clustering of the appropriate cell line samples (Figure 6a). NALM-6 sample distribution exhibits a clear variance between the Ento-exposed and control cells, while the SEM and RS4;11 samples do not differ from each other. Using filter criteria of Avg (log2) ≥ 5, fold change ≥ 2 or ≤ 2 within Transcriptome Analysis Console (TAC) software, we identified 1.13% relevantly up- and 10.73% downregulated genes in the pre-B-ALL cell line NALM-6. We found 2 up- and 19 downregulated genes within the BCR/PI3K/AKT pathway following Ento exposure (Figure 6c). *PTPN6* is the most downregulated gene in Ento-exposed NALM-6 cells compared to the DMSO control (fold change: −10.99), followed by *BCL-6* (fold change: −9.62). Furthermore, we detected the downregulation *of LAT2, IL7R, DAPP1, CD22* and more. Ento administration only upregulated a minor set of some BCR genes like *AKT3* and *RPS6KB1* in NALM-6. However, the pro-B-ALL SEM showed only 1.13% up- and 1.13% downregulated genes (Figure 6a,b,d,e). *CD79B* (fold change: 2.41) and *FOXO1* (fold change 2.04) were upregulated, whereas *VEGFA* (fold change: −2.01) and *SPRY2* (fold change: −2.01) were downregulated after Ento exposure. In contrast, in the pro-B-ALL RS4;11, no relevant gene expression changes could be identified after Ento mono exposure during RNA panel sequencing. All read counts and normalized read counts are shown in Appendix A.

### 2.8. Downstream Protein Characterization Unveiled p53 and BCL-6 as Key Targets by Entospletinib Exposure

In the pre-B-ALL cell line NALM-6 no phosphorylation of SYK_Tyr525/526_ was detected, while SYK_Y352_ slightly increased during the tested time points (Figure 7c). Total SYK was not affected. pERK and total ERK expression remained mostly unaffected. A slight concentration-dependent decrease in p4E-BP1 was observed in NALM-6, whereas p53 was not detectable. pSHP-1 and total SHP-1 were only detectable starting from 48 h (Appendix A, Appendix A). Both decreased with increasing Ento concentrations. Further, pGSK3β considerably increased after 1 µM Ento exposure, but not in the higher concentrations. Total GSK3β remained unaffected. Interestingly, BCL-6 was detected in the DMSO control and decreased after Ento exposure, independently of the concentration. Further, a strong pAKT level was detected after 1 µM Ento exposure, which then declined with higher concentrations again. The appropriate total AKT expression was consistent. Complete Western blot analyses for 24 h, 48 h and 72 h Ento exposure on NALM-6 cells are shown in Appendix A (Appendix A).

Phosphorylated SYK decreased concentration dependent in the pro-B-ALL cell line SEM, validated for both phosphorylation residues Tyr525/526 and Y352 (Figure 8 and Appendix A, Appendix A). Application of 10 µM Ento almost completely inhibited the pSYK expression, while total SYK remained unaffected. Interestingly, pERK expression increased by Ento exposure in SEM, while total ERK remained unaffected. Strikingly, the tumor suppressor p53 vigorously decreased concentration dependently in SEM. Furthermore, the phosphorylated SHP-1 decreased only during the first 24 h and pGSK3β increased independently of concentration and time. Total GSK3β was not influenced. Furthermore, BCL-6 protein expression has not been detected in SEM during the tested time points. Another interesting modification has been observed in the amount of phosphorylation of AKT residue Ser473. In SEM cells, pAKT was increased in all samples after Ento exposure compared to the DMSO control. Distinct changes were observed after long-term incubation (72 h), whereby 1 µM Ento exposure induced distinct modification merely after 24 h and 48 h. Total AKT expression remained unaffected during all time points. Complete Western blot analyses for 24 h, 48 h and 72 h Ento exposure on SEM cells are shown in Appendix A (Appendix A).

In the second pro-B-ALL cell line RS4;11, an increase of SYK_Y352_ level was detected during the tested time points while total SYK remained unaffected (Figure 9 and Appendix A, Appendix A). pERK expression was distinctly reduced after Ento exposure during all tested time points, while total ERK was not affected. Further, no p53 expression was detected during the tested conditions. pSHP-1 results were discontinuous and showed an increase only after 1 µM Ento exposure during 72 h incubation, while total SHP-1 was not affected. Moreover, high level of BCL-6 was detected in all samples and time points without any alteration. No obvious expression changes were observed in pGSK3β and total GSK3β, pAKT and total AKT during the tested conditions. Complete western blot analyses for 24 h, 48 h and 72 h Ento exposure on RS4;11 cells are shown in Appendix A (Appendix A).

Moreover, Western blot densitometric analyses of the NALM-6 exposed cells confirmed significant changes in BCL-6, pSHP-1, pGSK3β and pAKT expression (Figure 7d). Further distinct changes were observed in SYK_Y352_ and p4E-BP1 expression. Pro-B-ALL SEM Western blot quantification also verified the most vigorous and significant alteration of SYK_Tyr525/526_, SYK_Y352_ and p53. Further modifications were observed in pERK, pGSK3β and pAKT expression. Exemplarily, the densitometric analysis of the mostly affected proteins are shown in Figure 8b. The quantification of Ento-exposed pro-B-ALL cell line RS4;11 revealed distinct modifications in SYK_Y352_ and pERK expression (Figure 9b).

## 3. Discussion

Targeting B-cell receptor associated kinases, such as SYK, represents a major therapeutic tool in the treatment of B-cell malignancies such as CLL and DLBCL. In progenitor and precursor B-ALL, the function and druggability of SYK is still controversially discussed. Here, we identified a diverse acting mode of entospletinib, which not only relies on pre-BCR dependency. Anti-proliferative and pro-apoptotic effects were observed in both pre-B-ALL (NALM-6) and pro-B-ALL (SEM) cell lines, while a second pro-B-ALL (RS4;11) remained resistant, independently of the applied Ento concentration. Currently, BCR dependency is considered as a response marker to SYK inhibition. Here, pre-BCR^+^ and pre-BCR^−^ ALL cells can be distinguished by kinase inhibitor sensitivity, where only pre-BCR^+^ ALL cells selectively respond to SYK inhibition [27]. However, in this study we describe an exception for pro-B-ALL cells, showing a comparable sensitivity to SYK inhibition as do the pre-B-ALL cells, with comparable Ento IC50 values. Further, the observed pro-apoptotic effects were slightly more pronounced in the pro-B-ALL cells (SEM) compared to the pre-B-ALL cells (NALM-6). In contrast, apoptosis induction was not observed in the second pro-B-ALL cell line RS4;11. Compared to other studies testing Ento in CLL in vitro, the anti-proliferative as well as pro-apoptotic effects were slightly higher compared to our tested B-ALL cell lines [25,28]. CLL cells depends on their microenvironment, cell homing, chemokine/cytokine interactions and activation of B-cell receptor signaling [29]. Disrupting BCR signaling by BCR-associated kinase inhibition, e.g., BTK or SYK, lead to cross-talk blockage within the microenvironment and thus of BCR signaling [30], which makes SYK inhibition in CLL highly effective. In accordance to that, strong apoptosis induction in sensitive DLBCL cell lines by SYK inhibitor R406 was also shown and reasoned by intact BCR signaling [14]. Presumably, CLL and DLBCL depend more on functional BCR signaling compared to precursor B-ALL and progenitor B-ALL in any case, due to the lacking BCR or pre-BCR. Depending on whether a pre-BCR is present or not, it can be assumed that the acting mode of SYK inhibition and the followed signal transduction will differ from each other. Cell-cycle distribution of pre-B-ALL cells (NALM-6) and pro-B-ALL cells (SEM and RS4;11) were not affected by Ento treatment. This is in contrast to the findings in DLBCL cell lines to SYK inhibition by PRT318. In this study, SYK inhibition induced cell-cycle arrest by blocking G1/S transition in a part of the sensitive DLBCL cell lines [31]. Moreover, SYK itself is known to be a master regulator of anti-apoptotic signaling in B-lineage leukemias and lymphomas with cell-cycle regulatory function by G2 checkpoint regulation. SYK occurs to be essential for cell-cycle control due to CDC25C phosphatase inactivation by phosphorylation of S216 residues and further maintaining CDK1 inactive. This prevents CDK1 entry into mitosis and encourages SYK as an important regulator during the cell-cycle process [32]. In B-ALL cell lines, cell-cycle distribution does not seem to be influenced by SYK inhibition.

We further investigated the Ento-induced effects on BCR and PI3K/AKT downstream signaling by Western blot analyses. Here, we focused on the pivotal kinases. All tested cell lines exhibit different pathway modulation after Ento exposure. Whereas RS4;11 protein expression was only moderately modified by Ento treatment, SEM and NALM-6 were markedly affected. The most conspicuous alterations in the pro-B-ALL cell line SEM was the reduction of pSYK_Tyr525/526_ and p53 and the increase of pERK1/2. It is already known, that SYK is able to modulate p53 activity [33,34]. p53 (*TP53* gene) normally bears a tumor-suppressor function, where a high p53 induces apoptosis, cell-cycle arrest and senescence [35]. In cancers, *TP53* is frequently mutated [36], resulting in loss-of-function (LoF), dominant-negative (DN) or gain-of-function (GoF) effects. *TP53* hot-spot mutants, as R157H, R282W, R248Q and R249, can destabilize the DNA-binding domain, which can lead to partially unfolded p53 proteins and therefore loss of the p53 wild type (wt) function [37]. The substitution in R248Q also causes structural alteration in the respective DNA contact sites [38,39]. Further, R248Q, have been shown to build aggregates caused by protein misfolding [38,39]. Aggregated p53 mutants inactivate p53 and leads to LoF effects. Moreover, aggregated p53 forms can also co-aggregate with other tumor suppressors, such as p63 or p73. This interaction with new DNA-binding sites can explain the GoF effects that, in turn, increases cancer progression [38].

The pro-B-ALL cell line SEM carries the *TP53* p.R248Q mutation [40] (COSM10662), which could explain the high p53 expression of the stabilized p53 mutant. Ento was able to reduce p53 in the pro-B-ALL cell line SEM as a consequence of the pSYK decrease.

In pre-B-ALL NALM-6, especially 1µM Ento induced obvious changes within BCR and PI3K/AKT signaling. Here, we found pAKT and pGSK3β upregulated by Ento exposure. Moreover, in DLBCL cell lines, SYK inhibition by R406 reduced the downstream signaling of the BCR by pAKT, pGSK3β and pERK reduction [41]. In contrast, Ento exposure on pro-B-ALL SEM increased the pERK1/2 level concentration independently. Previously, it was demonstrated that Raf/MEK/ERK activation can induce drug resistance. Moreover, a decrease in p53 was associated with drug resistance [42]. The pAKT and pGSK3β levels were also increased in SEM after Ento exposure. These distinct modifications in SEM seem to indicate a drug-escape mechanism caused by entospletinib exposure. In acute myeloid leukemia (AML), an activated RAS/MAPK/ERK pathway also was observed after Ento exposure and linked to resistance to Ento [43,44].

The reduction in PTPN6 and pSHP-1 in NALM-6 in response to SYK inhibition presumably occurs due to a negative feedback loop based on a decreased pSYK level. PTPN6, also known as SHP-1, is a downstream protein tyrosine phosphatase of SYK that negatively regulates SYK and attenuates BCR signaling [45,46]. Whereas SYK is important for B-cell development, SHP-1 is a negative regulator of B lymphocyte activation [47] and counteracts the phosphorylation of SYK [48].

However, Western blot analyses revealed a downregulation of BCL-6 and pSHP-1 in NALM-6 after Ento exposure. This observation was also confirmed by RNA panel sequencing, where *BCL-6* and *PTPN6* were likewise mostly downregulated. It was already shown that BCL-6 directly depends on pre-BCR signaling, where SYK inhibition reduced BCL-6 in the *TCF3-PBX1* ALL cell line and patient-derived *TCF3-PBX1* ALL cells [27]. Further, Geng et al. demonstrated a SYK-dependent BCL-6 expression in pre-BCR^+^ ALL cells and with that a poor outcome for high BCL-6 expression levels in pre-BCR^+^ ALL [27]. It was concluded that BCL-6 expression depends on tonic pre-BCR signaling due to BCL-6 repression by SYK and SRC inhibition [27]. However, we found BCL-6 highly expressed in the pro-B-ALL cell line RS4;11, at the RNA and protein level, where Ento exposure did not affect the BCL-6 expression nor cell viability. This could suggest that, independent of pre-BCR signaling, high BCL-6 expression prevents the cell from apoptosis induction due to the BCL-6–BIM axis. This mechanism was already proved in a recent study, where BCL-6 function was elucidated in *MLL*-rearranged B-ALL. Initially, it was shown that especially *MLL*-rearranged B-ALL showed an upregulated BCL-6 expression, which is also related to a poor prognosis. Moreover, it was shown that BCL-6 is able to bind the BCL2L11 locus (BIM) and consequently regulates BIM expression. The deletion of *BCL-6* induced an increased BIM level [49]. Altogether, a high BCL-6 expression keeps pro-apoptotic BIM under control, which in turn prevents apoptosis induction in *MLL*-rearranged B-ALL. Low BIM expression or BIM loss correlates with a pro-survival behavior and was already observed in many cancers as a survival strategy [50,51]. For that reason, the low BCL-6 expression in *MLL*-rearranged sensitive SEM cells, high BCL-6 expression in *MLL*-rearranged resistant RS4;11 and the reduction of BCL-6 expression in NALM-6 could be explained by the BCL-6–BIM axis and with that the sensitivity to Ento exposure. Concerning this, a new study from Toscan et al. revealed a strategy to overcome glucocorticoid resistance in ALL due to recovering the BIM function [52].

Altogether, our present study demonstrates anti-proliferative, metabolism modulating and pro-apoptotic effects in pre- and pro-B-ALL cell lines in vitro, independent of pre-BCR presence. We further unveiled that the *TP53* variant in pro-B-ALL SEM and the BCL-6–BIM axis can probably explain the entospletinib sensitivity and therefor provides a rationale for Ento application also in a subset of pro-B-ALL.

## 4. Materials and Methods

### 4.1. Cell Lines and Reagents

Human B-ALL pre-B-ALL cell line NALM-6 (pre-BCR^+^), pro-B-ALL cell lines SEM and RS4;11 (pre-BCR^−^), B lymphoma cell line SU-DHL-4 and T lymphoma cell line SUP-T1 were purchased from the German Collection of Microorganisms and Cell Cultures (DSMZ, Braunschweig, Germany). The cells were cultured as recommended by the manufacturer at 37 °C and 5% CO_2_ in media supplemented with 10 % heat-inactivated FCS (Biochrom, Berlin, Germany) and 100 µg/mL penicillin and streptomycin (Biochrom, Berlin, Germany). Mycoplasma contamination was regularly tested.

Entospletinib (GS-9973) was purchased from Selleck Chemicals (Absource Diagnostics GmbH, München, Germany, catalog no. S7523). A 10 mM Stock solution was prepared according to the manufacturer’s instructions and stored at −80 °C.

### 4.2. Drug Exposure Experiments

B-ALL cell lines (3.3 × 10^5^ cells/mL) were treated with serial end-concentrations (0.001–10 μM) of entospletinib. Cells were cultured in the appropriate medium containing 0.1% (*v*/*v*) DMSO as the solvent of the drug or dose ranges of entospletinib as single substance for 24 h, 48 h and 72 h. Following, the effect on cell proliferation (trypan blue staining), metabolism (WST-1 assay), apoptosis/necrosis (annexin V/PI staining), cell morphology and protein expression were determined. All experiments were performed at least in three biological replicates.

### 4.3. Cell Proliferation

Pro-B-ALL cells SEM and RS4;11 as well as pre-B-ALL cell line NALM-6 were seeded at a density of 5 × 10^5^ cells in 1.5 mL in 24-well plates as duplets (Greiner Bio-One GmbH, Frickenhausen, Germany). The B-ALL cell lines were exposed to the entospletinib or 0.1% (*v*/*v*) DMSO as control. The cells were harvested 24 h, 48 h and 72 h after drug incubation and washed with PBS (Biochrom, Berlin, Germany), (10 min, 180 g, 4 °C). The number of viable cells was determined by cell count with a hemocytometer after trypan blue staining (Sigma-Aldrich Chemie GmbH, Steinheim, Germany).

### 4.4. WST-1 Proliferation Assay

Cells were seeded at a density of 5 × 10^4^ cells/well in 96-well plates with 150 µL cell culture medium containing DMSO as the control or entospletinib, in triplicate. Evaluation of the metabolic activity was analyzed via tetrazolium compound WST-1 (TaKaRa Bio Inc., Kusatsu, Japan, catalog no. MK400) according to the manufacturer’s protocol. Briefly, after an appropriate incubation time, the cells were incubated with 15 µL WST-1 premix WST-1 for up to 3 h. The absorbance at 450 nm and a reference wavelength at 750 nm were determined using the GloMax- Multi Microplate Multimode Reader (Promega, Madison, WI, USA). The absorbance of only culture medium containing the WST-1 reagent was used as the background control.

### 4.5. Apoptosis Assay

Early and late apoptosis was analyzed by annexin V FITC (BD Biosciences, Heidelberg, Germany, catalog no. 556419) and propidium iodide (PI) (Sigma Aldrich, St. Louis, MO, USA, catalog no. P4864-10ML) staining according to the manufacturer’s protocol. Drug-exposed and control cells were harvest from 24-well plates and washed twice with cold PBS (10 min, 180 g, 4 °C). The cell pellets were resuspended in a 100 µL annexin binding buffer (BD Biosciences, Heidelberg, Germany, catalog no. 556454) and stained with 5 µL annexin V FITC for 15 min at room temperature in the dark. Cell suspensions were adjusted to a final volume of 500 μL with the annexin binding buffer and stained with PI (20 μg/mL) immediately before measurement. Unstained and single stained control cells were included in each experiment. Measurements were performed using FACS Verse^TM^ (BD Biosciences, Heidelberg, Germany) and BD FACSuite Software (Version 4.0.2, BD Biosciences, Heidelberg, Germany).

### 4.6. Cell-Cycle Analysis

The DNA intercalating dye propidium iodide (PI) was used for cell-cycle analysis to quantify the DNA content. Cells were treated with 1 µM and 10 µM entospletinib or DMSO as the vehicle. After 48 h and 72 h at 37 °C entospletinib exposure, cells were harvested. After centrifugation at 180× *g* for 10 min at 4 °C the cells were washed twice with PBS, followed by fixing the cells with 70 % ice cold ethanol and a freezing cycle at −20 °C (≥24 h). Subsequently, the fixation solution was removed by centrifugation at 180× *g* for 10 min at 4 °C. The fixed cells were then washed twice with PBS. After centrifugation the cell pellet was then resuspended in 500 µL ribonuclease A (1 mg/mL) (Carl Roth, Karlsruhe, Germany, catalog no. 7156.2). RNase treatment ensued 45 min at 37 °C. After a centrifugation step at 180× *g* for 10 min at 4 °C, the cells were washed again with PBS for two times. PBS was removed by a centrifugation, the pellet was resuspended in 400 µL PI Solution (50 µg/mL) (Sigma-Aldrich Chemie GmbH, Steinheim, Germany, catalog no. P4864-10ML) and analyzed by flow cytometry (FACS Verse^TM^, BD Biosciences, Heidelberg, Germany) and BD FACSuite Software (Version 4.0.2, BD Biosciences, Heidelberg, Germany).

### 4.7. Immunofluorescence Staining

Coated cytoslides (Tharmac, Waldsolms, Germany, catalog no. JC311-100) were prepared with 5 × 10^4^ cells per slide (750 rpm, 10 min) in a Cytospin 3 centrifuge (Shandon, Frankfurt/Main, Germany). Slides were fixed in ice-cold methanol for 10 min. Afterward, the cells were permeabilized, blocked and stained with SYK antibody (1:400 diluted in 1% BSA) (sc-1240, Santa Cruz Biotechnology, Santa Cruz, TX, USA). Slides were washed and stained with secondary goat Anti-mouse IgG Alexa Fluor Plus 488 (1:1000 diluted in 1% BSA) (Thermo Fisher Scientific, MA, USA). The antibodies used are listed in Appendix A, Appendix A. Finally, the cells were mounted, and the cell nuclei stained with Roti-Mount FluorCare DAPI (Carl Roth, Karlsruhe, Germany, catalog no. HP20.1). Fluorescence images were captured using a Nikon Eclipse TE2000-E confocal laser scanning microscope (Alexa Fluor 488 for SYK and FluorCare DAPI for nucleic staining (Carl Roth, Karlsruhe, Germany), with a 60× water-immersion objective and the EZ-C1 software program (Nikon, Düsseldorf, Germany, Version 3.80). Fluorescence images were processed with the ImageJ/Fiji software (Version 1.49b / Java 1.6.0_24 (64bit) [53,54].

### 4.8. Pappenheim Staining

Coated cytoslides (Tharmac, Waldsolms, Germany, catalog no. JC311-100) were prepared with 5 × 10^4^ cells per slide (750 rpm, 10 min) in a Cytospin 3 centrifuge (Shandon, Frankfurt/Main, Germany). The slides were stained for 6 min with May-Grünwald working solution (Merck, Darmstadt, Germany, catalog no. 101424), rinsed with microscopy buffer according to WEISE (pH7.2) (Merck, Darmstadt, Germany, catalog no. 109468), stained for 20 min in a Giemsa working solution (Merck, Darmstadt, Germany, catalog no. 109204) and rinsed thoroughly with the microscopy buffer again. The slides were air dried at room temperature before analysis. To evaluate the morphological changes of the cells, the slides were analyzed by EVOS^®^ XL Core Imaging System (AMG, Washington, DC, USA).

### 4.9. Protein Extraction and Western Blot Analyses

Treated cells were lysed using RIPA buffer (10 × RIPA Buffer; 100 × Protease/Phosphatase Inhibitor Cocktail (Cell Signaling Technology, Danvers, MA, USA, catalog no. #9806; catalog no. #5872)) and ultrasound. Protein concentrations were determined by Bradford Protein Assay (Bio-Rad, München, Germany). For each sample, 30 µg protein were separated on Criterion TGX Precast Gels (Bio-Rad, München, Germany, catalog no. 5671123). They were blotted onto a PVDF membrane (Trans-Blot Turbo Transfer Pack 0.2 µM PVDF, Bio-Rad, catalog no. 1704157) using Trans-Blot^®^ Turbo^TM^ Transfer System (Bio-Rad, 2.5 A, 25 V, 10 min). After blockage in 1:3 diluted Odyssey Blocking Buffer (LI-COR, Nebraska, USA, catalog no. 927-50000) for one hour, the membranes were incubated with the primary antibody overnight at 4 °C. The antibodies used are listed in Appendix A, Appendix A. The membranes were then washed in PBS-T and incubated with the secondary antibody for one hour. After a washing step in PBS-T the detection was performed with a LI-COR Odyssey Imaging System and Image Studio Lite software (LI-COR, Lincoln, NE, USA).

### 4.10. Intracellular Phospho-Specific Flow Cytometry and Stimulation

The detection of the intracellular targets pSYK and SYK was carried out by intracellular flow cytometry (FACS Verse^TM^, BD Biosciences, Heidelberg, Germany). The SUP-T1 T-cell lymphoma cell line acted as the negative control due to lacking SYK expression. The IgG-stimulated B-lymphoma cell line SU-DHL-4 acted as the positive control. AffiniPure F(ab’)_2_ Fragment Goat Anti-Human IgG, F(ab’)_2_ fragment specific was used for stimulation and purchased from Dianova (Jackson ImmunoResearch, West Grove, PA, USA, catalog no. #109-006-097). Here, SU-DHL-4 was seeded in 24-well plates (10^6^ cells/mL) and stimulated for 15 min with 5 µg/mL IgG at 37 °C and 5 % CO_2_ [9]. Cells were harvested directly from cell culture flasks and washed twice with 2 mL PBS. Cells were fixed with 4% formaldehyde (16%, methanol-free, Ultra-Pure EM Grade, Polysciences, Warrington, PA, USA, catalog no. 18814-10) to a final concentration of 4 % for 15 min at room temperature. Subsequently, cells were washed with an excess of PBS and resuspended in an appropriate amount of PBS. For permeabilizing the cells, ice-cold methanol was gently added to a final concentration of 90%. The cells were then incubated for 30 min on ice. Staining was started by washing the cells with an excess of PBS (300 g, 10 min, 4 °C). Cells were resuspended in incubation buffer (0.5% BSA in PBS) containing the antibody to a total volume of 100 µL. The conjugated antibodies used are listed in Appendix A, Appendix A. Cells were then incubated for 60 min in the dark at room temperature. Thereafter, stained cells were washed twice with PBS, resuspended in 300 µL PBS and were analyzed by flow cytometry (FACS Verse^TM^, BD Biosciences, Heidelberg, Germany). For all experiments, the unstimulated, autofluorescence and isotype controls were analyzed simultaneously. All experiments were performed at least in three biological replicates.

### 4.11. RNA Extraction and Isolation

Treated cells or control cells were grown in 6-well plates for up to 72 h, harvested and washed three times with cold PBS by centrifugation steps at 180× *g* for 8 min. Cell pellets were resuspended in QIAzol Lysis Reagent (QIAGEN, Venlo, Netherlands, catalog no. 79306) and RNA Isolation was performed with miRNeasy Mini Kit (QIAGEN, Venlo, Netherlands, catalog no. 217004) according to the manufacturer’s instructions. In brief, after lysis, the aqueous phase that contains the total RNA of the lysed cells were extracted and purified by a silica membrane of RNeasy Mini spin columns. The remaining DNA was digested by RNase-free DNase (QIAGEN, Venlo, Netherlands, catalog no. 79256) and the contaminants were washed away. Total RNA was eluted by nuclease-free water and RNA concentration measured on a Nanodrop Spectrophotometer ND1000 (PEQLAB Biotechnologie GmbH, Erlangen, Germany, Version 3.7.1).

### 4.12. Targeted RNA Sequencing and Data Analysis

Expression analyses were performed with AmpliSeq RNA targeted sequencing on the Ion Personal Genome Machine™ (PGM™) System (Ion Torrent™) (Thermo Fisher Scientific, Waltham, Massachusetts, USA). For targeted RNA sequencing, an in-house custom panel was designed with the Ion AmpliSeq Designer (Thermo Fisher Scientific), containing 179 genes mainly involved in B-cell-receptor and PI3K/AKT pathway signaling and more. Ion AmpliSeq RNA Libraries were prepared according to the manufacturer’s protocol (MAN0006735). In brief, RNA was quantified with the Qubit RNA HS Assay Kit (Thermo Fisher Scientific, Waltham, MA, USA, catalog no. Q32852) on the Qubit 2.0 Fluorometer (Life Technologies, Carlsbad, CA, USA) and transcribed into cDNA using SuperScript VILO cDNA Synthesis Kit (Thermo Fisher Scientific, catalog no. 11754-050). cDNA targets were amplified, amplicons partially digested, ligated to the adapters and purified with the Ion AmpliSeq™ Library Kit 2.0 (Thermo Fisher Scientific, Waltham, MA, USA, catalog no. 4475345). All steps were performed in the ProFlex PCR System (Thermo Fisher Scientific, Waltham, MA, USA). The final library was quantified by the Ion Library TaqMan Quantification Kit (Thermo Fisher Scientific, Waltham, MA, USA, catalog no. 4468802) with the ViiA 7 Real-Time PCR System (Thermo Fisher Scientific, Waltham, MA, USA). Thereafter, template preparation was carried out with the Ion PGM Hi-Q View OT2 Kit—200 in the Ion One Touch 2 Instrument (Thermo Fisher Scientific, Waltham, MA, USA, catalog no. A29900) and enrichment of template-positive Ion Sphere Particles (ISPs) with the Ion OneTouch ES (Thermo Fisher Scientific, Waltham, MA, USA, catalog no. 4478525). Finally, the sequencing run was performed with the PGM System. The evaluations of the data sets were performed using Transcriptome Analysis Console (TAC) Software (Thermo Fisher Scientific, Waltham, MA, USA, Version 4.0.0.25). The following filter criteria were set to identify relevantly regulated genes: Avg (log2) > 5, fold change > 2 or < −2.

### 4.13. Statistics and Reproducibility

Statistical data are presented as the mean ± standard deviation (SD) from at least three biologically independent samples using GraphPad Prism Software (San Diego, CA, USA, Version 8.0.2). Statistical differences between multiple comparisons were analyzed using one-way ANOVA followed by Dunnett’s multiple comparison test as a post-hoc analysis. Differences were considered statistically significant for * *p* < 0.05, ** *p* < 0.001, *** *p* < 0.0001.

## 5. Conclusions

In this study, we provided evidence for a potential anti-leukemic effect of entospletinib in pre- and pro-B-ALL cell lines in vitro, independent of their pre-BCR status. Opposed to the expectation, the pro-B-ALL cell line SEM showed a good Ento response, such as anti-proliferative, metabolism and pro-apoptotic effects. We identified a diverse acting mode of Ento, independent of pre-BCR dependency, based on gene and protein modulation after Ento exposure. Here, the *TP53* variant in pro-B-ALL SEM and the BCL-6–BIM axis are probably Ento key targets and can explain the Ento sensitivity, therefor providing a rationale for Ento application also in a subset of pro-B-ALL.

## Figures and Tables

**Figure 1 ijms-22-00592-f001:**
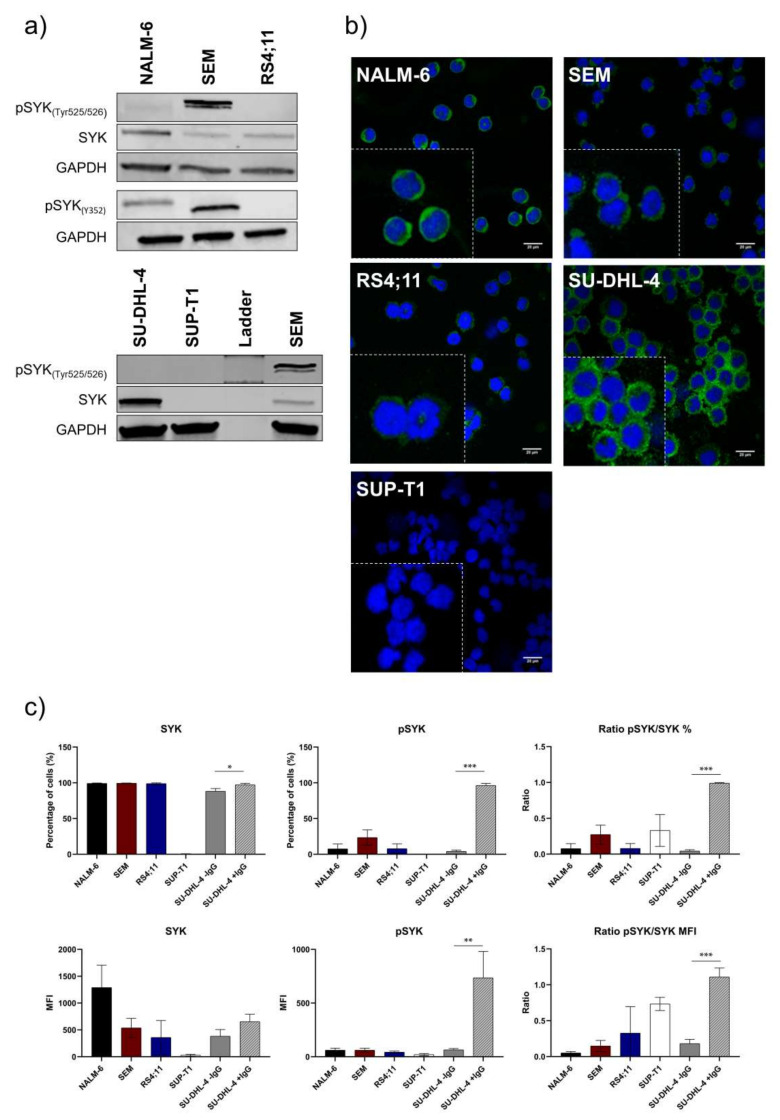
Basal characterization of SYK and pSYK expression in B-ALL cell lines. Characterization of basal SYK protein expression in B-ALL cell lines. Highest SYK expression was observed in pre-B-ALL NALM-6, while pro-B-ALL SEM showed the highest pSYK expression. (**a**) SYK and pSYK expression of two phosphorylation sites (Tyr525/526 and Y352) under non-stimulating conditions in the B-ALL cell lines and the DLBCL cell line SU-DHL-4 as well as the T-cell lymphoma cell line SUP-T1 (positive and negative control, respectively). (**b**) Immunofluorescent staining of SYK in the B-ALL cell lines and control cell lines (composite image of the Alexa Fluor Plus 488 green-labeled SYK and blue DAPI-stained nucleus) by confocal microscopy. (**c**) Intracellular staining of SYK-FITC and pSYK(Y348)-PE by flow cytometry in unstimulated B-ALL cells showing SYK and pSYK expression based on percentage (%) and mean fluorescent intensity (MFI) (SUP-T1 cell line as the negative control and IgG-stimulated SU-DHL-4 as the positive control). The mean fluorescence intensity (MFI) of all cell lines is expressed in arbitrary units (AU). Data are presented as the mean ± SD. Statistical significance was calculated by *t*-test and displayed as * *p* < 0.033, ** *p* < 0.002, *** *p* < 0.001 (*n* ≥ 3).

**Figure 2 ijms-22-00592-f002:**
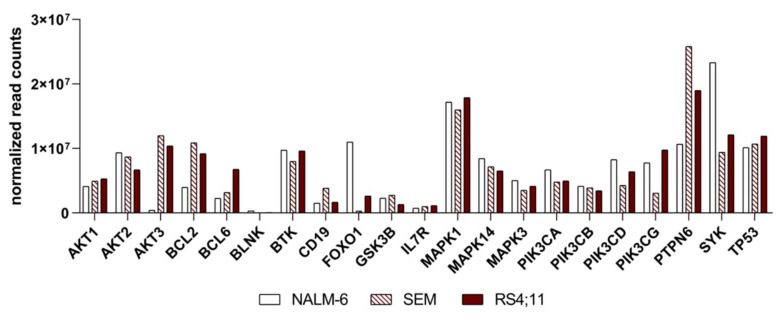
Basal expression of the B-cell receptor and PI3K/AKT pathway genes. Representation of the BCR and PI3K/AKT pathway key regulators in all B-ALL cell lines. Basal expression of selected essential BCR and PI3K/AKT pathway genes; targeted sequencing normalized read counts.

**Figure 3 ijms-22-00592-f003:**
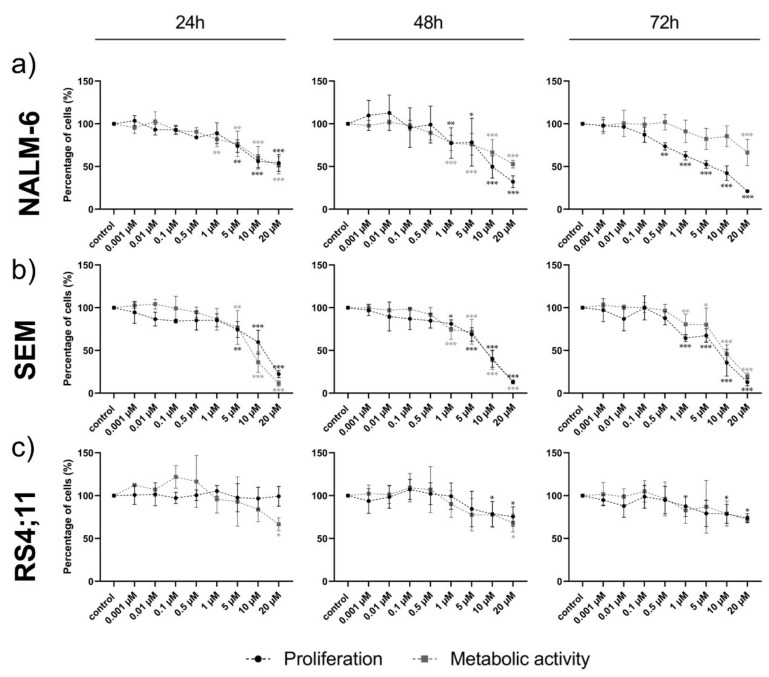
Cell viability after entospletinib exposure in B-ALL cell lines. Time- and concentration-dependent reduction in cell proliferation and metabolic activity in the pre-B-ALL cell line NALM-6 and pro-B-ALL cell lines SEM and RS4;11 by entospletinib. Cell proliferation and metabolic activity for entospletinib serially diluted at different concentrations (0.001 µM–20 µM) for 24 h, 48 h and 72 h (**a**) NALM-6, (**b**) SEM and (**c**) RS4;11. Data are presented as the mean ± SD. Statistical significance was calculated by one-way ANOVA followed by Dunnett’s multiple comparison test as a post-hoc analysis and displayed as * *p* < 0.033, ** *p* < 0.002, *** *p* < 0.001 versus the control group (*n* ≥ 3).

**Figure 4 ijms-22-00592-f004:**
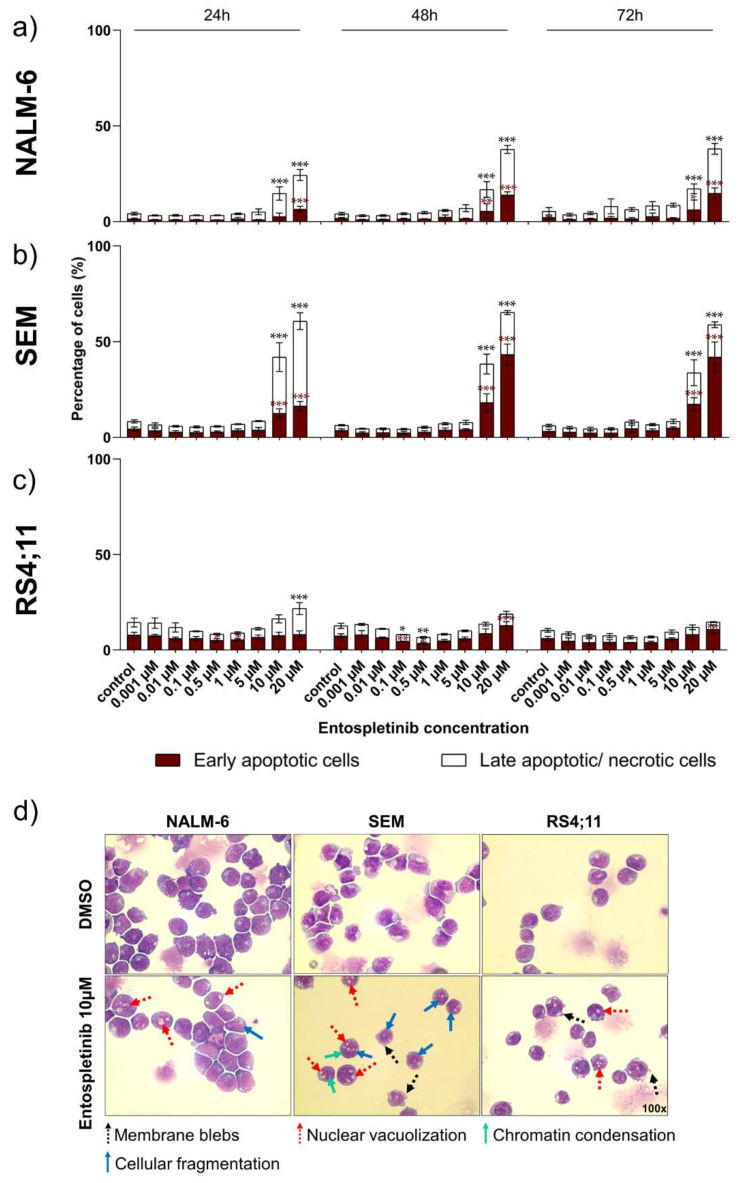
Apoptosis induction in B-ALL after entospletinib (Ento) exposure. Concentration-dependent apoptosis induction by entospletinib in pre-B-ALL NALM-6 and pro-B-ALL SEM. B-ALL cells were exposed to serially diluted entospletinib concentrations (0,001 µM–20 µM). Apoptosis induction was determined by Annexin V/PI staining for an exposure time of 24 h, 48 h and 72 h. (**a**) NALM-6 cells, (**b**) SEM and (**c**) RS4;11, respectively. (**d**) Representative light microscopy images (×100) of entospletinib-exposed B-ALL cells after 72 h. Cytospins were stained with May-Gruenwald Giemsa stain (Pappenheim method) after 24 h, 48 h and 72 h entospletinib exposure at different concentrations (1 µM, 5 µM and 10 µM). Data are presented as the mean ± SD. Statistical significance was calculated by one-way ANOVA followed by Dunnett’s test as a post-hoc analysis and displayed as * *p* < 0.033, ** *p* < 0.002, *** *p* < 0.001 versus the control group (*n* ≥ 3). Black asterisks indicate significance of late apoptotic/necrotic cells and red asterisks indicate significance of early apoptotic cells.

**Figure 5 ijms-22-00592-f005:**
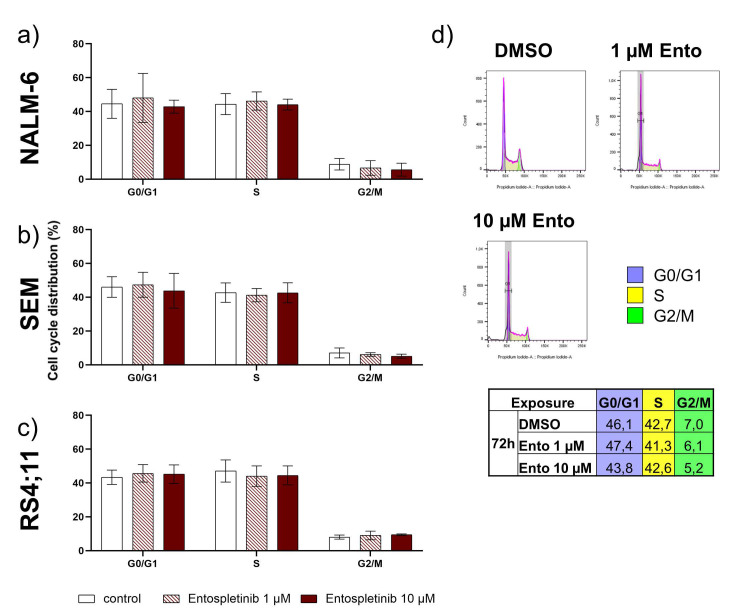
Flow cytometric cell-cycle analysis of the B-ALL cells after entospletinib exposure. B-ALL cell lines NALM-6, SEM and RS4;11 were exposed to 1 µM or 10 µM entospletinib for 72 h. (**a**) Mean cell-cycle distribution of NALM-6, (**b**) SEM and (**c**) RS4,11. (**d**) Representative histograms of the cell-cycle distribution of the SEM cells exposed to entospletinib or the vehicle, calculated by FlowJo Software, and the mean calculation of the cell-cycle phases. Data are presented as the mean ± SD. Statistical significance was calculated by one-way ANOVA followed by Dunnett’s test as a post-hoc (*n* ≥ 3).

**Figure 6 ijms-22-00592-f006:**
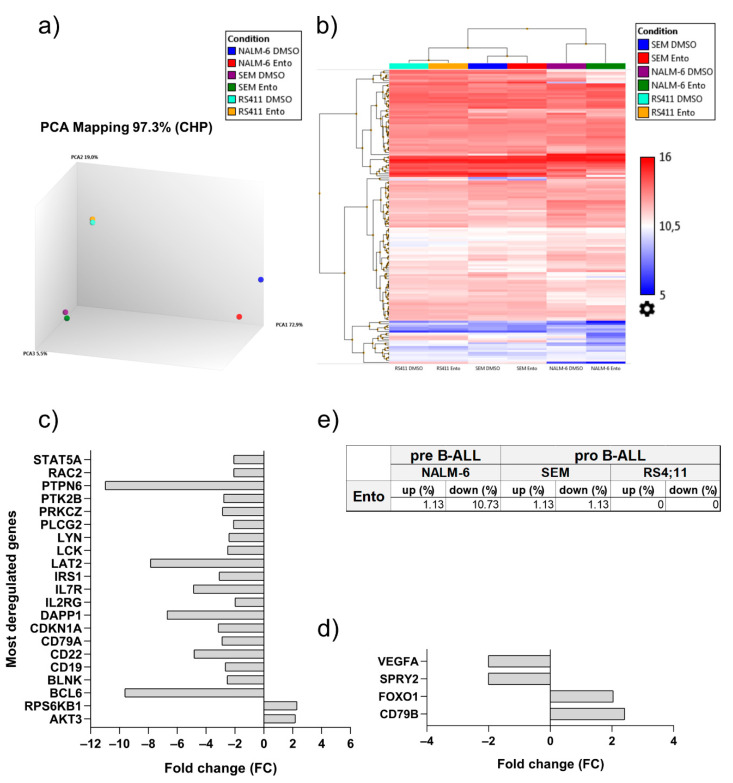
Entospletinib-induced gene expression changes in the B-ALL cell lines. Entospletinib-induced gene expression changes within the BCR and PI3K/AKT pathway predominantly in pre-B-ALL NALM-6 and slightly in pro-B-ALL SEM. (**a**) Principle component analysis of the control and 1 µM Ento-exposed B-ALL cell lines revealed distinct clustering of the appropriate cell line samples and variance in the NALM-6 cells after Ento exposure. (**b**) Hierarchical clustering of all the relevant regulated genes demonstrates the gene expression changes in the B-ALL cell lines. (**c**) Fold changes of the significantly deregulated genes in NALM-6 and (**d**) SEM. (**e**) Percentagewise deregulation after 1 µM Ento exposure.

**Figure 7 ijms-22-00592-f007:**
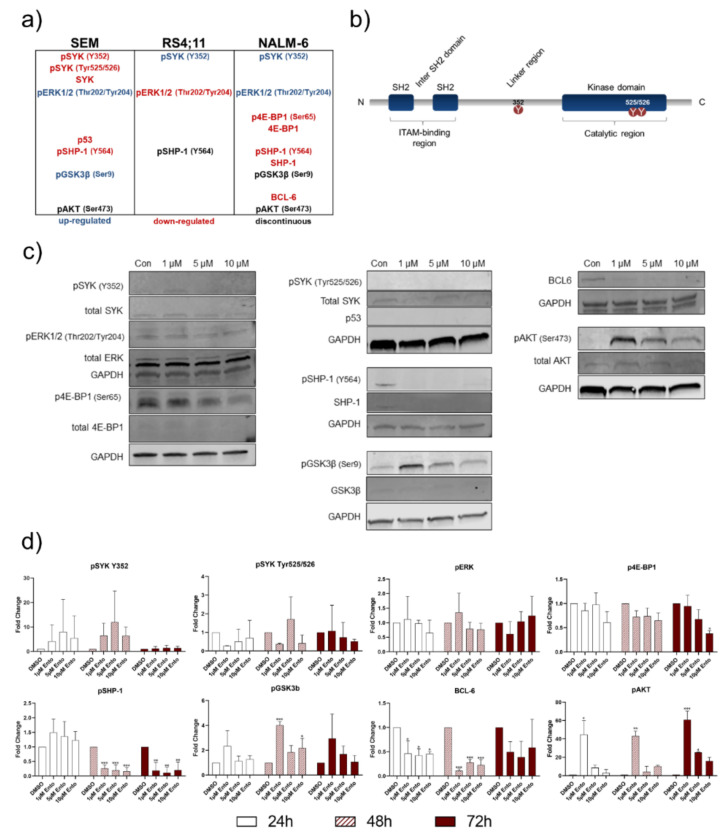
Entospletinib-induced distinct downstream protein modifications in pre-B-ALL NALM-6. Western Blot analyses revealed distinct changes in the SYK downstream proteins in pre-B-ALL NALM-6. (**a**) Condensed conclusion of protein modifications in all tested B-ALL cell lines. (**b**) Schematic illustration of the SYK protein structure, showing the amino acids used to examine the phosphorylation status. (**c**) Representative Western blot images of NALM-6 cells after 72 h entospletinib exposure. (**d**) Western blot quantification of key proteins after 24 h, 48 h and 72 h exposure (≥3 independent Western blots). Data are presented as the mean ± SD. Statistical significance was calculated by one-way ANOVA followed by Dunnett’s test as a post-hoc analysis and displayed as * *p* < 0.033, ** *p* < 0.002, *** *p* < 0.001 versus the control group (*n* ≥ 3).

**Figure 8 ijms-22-00592-f008:**
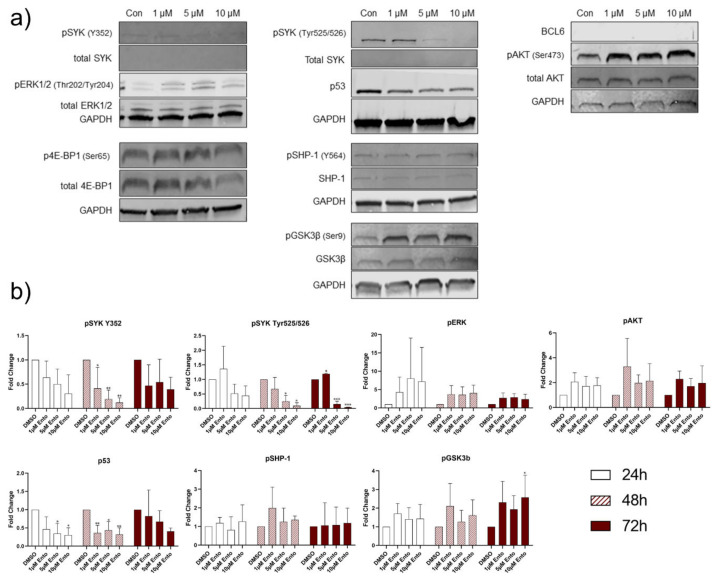
Entospletinib-induced distinct downstream protein modifications in pro-B-ALL SEM. (**a**) Representative Western blot images of SEM cells after 72 h exposure. (**b**) Western blot quantification of key proteins (≥3 independent Western blots for 24 h, 48 h and 72 h time points). Data are presented as the mean ± SD. Statistical significance was calculated by one-way ANOVA followed by Dunnett’s test as a post-hoc analysis and displayed as * *p* < 0.033, ** *p* < 0.002, *** *p* < 0.001 versus the control group (*n* ≥ 3).

**Figure 9 ijms-22-00592-f009:**
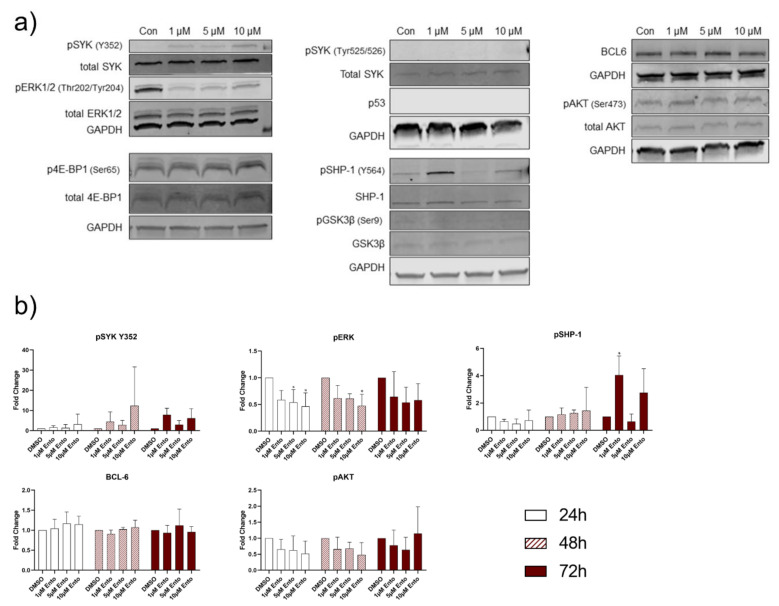
Entospletinib induced moderate downstream protein modifications in pro-B-ALL RS4;11. Entospletinib induces moderate protein expression changes in pro-B-ALL RS4;11. (**a**) Representative Western Blot images of RS4;11 cells after Ento exposure (**b**) Western Blot quantification of key proteins (≥ 3 independent Western Blots for 24 h, 48 h and 72 h time points). Data are presented as the mean ± SD. Statistical significance was calculated by One-way ANOVA followed by Dunnett’s test as post hoc (n ≥ 3).

## Data Availability

Data is contained within the article or Appendix A.

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
