# Peer review of "Precursor B-ALL Cell Lines Differentially Respond to SYK Inhibition by Entospletinib"

_ijms, 2021, doi:10.3390/ijms22020592_

Round 1

Reviewer 1 Report

In the present manuscript, Sender S. and co-workers evaluate the effect of Entospletinib (Ento) in a cohort of pre- and pro- B-ALL cell lines. The authors started the study evaluating the expression of the basal and activated forms of SYK kinase in three B-ALL cell lines, showing heterogeneous expression between the cell models. Then, authors showed the anti-proliferative and pro-apoptotic effects of Ento in NALM-6 and SEM but not in RS4;11 cells. Although the gene expression analysis of the B-ALL cells treated with Ento is properly design, the data regarding the in vitro effect of the compound in term of  induction of apoptosis and modification of cell cycle profile is poorly presented. Additional experiments are needed to sustain the presented results and to enforce the conclusions.

Major comments:

1) In the Result section Figure 1A-1B, authors evaluate the SYK kinase expression by immunoblots and immunofluorescence. However, immunoblots report only the data on the B-ALL cell lines and not the positive (SU-DHL-4) and negative (SUP-T1) controls. The authors should add SU-DHL-4 and SUP-T1 protein lysate in the immunoblots.

2) In the Result section Figure 4B, authors reported that in the SEM cells the treatment with Ento induces late apoptotic/necrotic death at 24 hrs and the early apoptotic death at 48 and 72 hrs. These results are very difficult to understand from a biological point of view. How do the authors explain this unusual mechanism of action of the compound? Moreover, the morphological analysis of Figure 4E do not provide any additional information regarding the mechanism of action of the compound.

3) In the Result section Figure 5, the cell cycle analysis is inaccurate. Authors reported the effect of the compound on NALM-6, RS4;11 and SEM cell lines, concluding that Ento at 1 or 10 uM induces G0/G1 cell cycle arrest after 48h of treatment in SEM cells. Based on the results reported on Figure 4, is hard to believe that the two drug concentrations have the same effect on cell cycle profile if you consider the effect of the same concentrations on apoptosis. Indeed, the first concentration (1uM) is almost not cytotoxic while the second once (10uM) induces around 40% of apoptosis. Moreover, it is hard to believe that in the control sample (white histograms) the percentage of cells in G0/G1 phase is lower than the percentage of cells in S phase. This is difficult to understand, not only from a biological point of view but also because the representative histograms of Figure 5D, nicely shows that in control cells (DMSO) the mathematical sum of the percentage of cells in the different phases is 79,7 and not 100%, around 20% of cells are missing.

Minor comments:

  • The quality of the dot blots of Figure 4D is very low. Authors should consider to improve picture quality or to remove the dot blots panels.
  • Again, the quality of figure is very poor and the differences between arrows (dashed or not) cannot be quantify.
  • In Figure 5 panel A, B and C the legend is missing.

Reviewer 2 Report

The authors used three pre- and pro-B ALL cell lines to test an effect of entospletinib as a potential anti-leukemic medicine. Overall, entospletinib resulted in reduced viability of SEM cells at relatively high doses (10-20 uM), and had a lower effect on NALM-6 cells. Viability of RS4;11 cells were nearly not affected even by high doses of the drug. The conclusions are based on 1-3 cell lines, and there is no validation using clinical material. This is a limitation of the study.

The manuscript is heavily loaded with data and would benefit from improved presentation and clarification of several points listed below.

Major:

1. Results section 2.6. Both data and conclusions do not appear justified enough. There is no cell cycle arrest, only a modest shift of %, which might be artificial. Only in SEM samples G0/G1 is significantly changed after application of the drug, and only after 48 hours. What is the evidence that S and G2/M % also changed significantly in this group (SEM, 48h)? All other groups presented in Figure 5a-c seems to have no cell cycle change after treatment with entospletinib. Why there is no effect at 72 h? Moreover, the authors claim that the trend was "marginally visible" in RS4;11. There is no data presented to justify this conclusion, e.g. statistics.

Minor:

2. Figure 1a (western blot) would benefit from a negative control where SYK is not detectable. It will confirm that the signal of lower intensity of SYK in SEM and RS4;11 cells is a specific band.

3. Images of Figure 2b are hardly visible.

4. Effect in this study is often observed using 10-20 uM of the drug. Is this concentration clinically relevant? What is known about potential concentrations used in human patients?

5. Figures 7, 8 and 9 are hardly readable/visible. The options could be to split the Figures into simpler ones, move some data to supplementary, or reformat the Figures. 

6. Materials and Methods. Nearly no catalog numbers of the products are presented. Kindly provide.

7. Conclusions. As for now, the part about "cell cycle modulating effect" is not justified enough.

Round 2

Reviewer 1 Report

No additional comments

Reviewer 2 Report

The authors provided a revised manuscript and addressed the questions raised by reviewers. Minor points:

  1. Section 2.6. "Entospletinib do not induces" is likely "Entospletinib does not induce"
  2. Lines 209-211. "Either low concentration of Ento (1 µM), or high concentration (10 µM) was able to induce cell cycle arrest in the B-ALL cell lines after 72 h incubation". Did the authors mean "was not able to induce". Given no changes in the cell cycle, this sentence is confusing as it is now. Kindly check
